# 1,25(OH)_2_D_3_ Promotes Macrophage Efferocytosis Partly by Upregulating *ASAP2* Transcription via the VDR-Bound Enhancer Region and ASAP2 May Affect Antiviral Immunity

**DOI:** 10.3390/nu14224935

**Published:** 2022-11-21

**Authors:** Hui Shi, Jiangling Duan, Jiayu Wang, Haohao Li, Zhiheng Wu, Shuaideng Wang, Xueyan Wu, Ming Lu

**Affiliations:** 1Department of Immunology, School of Basic Medical Sciences, Anhui Medical University, Hefei 230032, China; 2Department of Human Anatomy, School of Basic Medical Sciences, Anhui Medical University, Hefei 230032, China

**Keywords:** 1,25(OH)_2_D_3_, ASAP2, macrophages, efferocytosis, enhancer

## Abstract

The active form of vitamin D_3_, i.e., 1,25(OH)_2_D_3_, exerts an anti-inflammatory effect on the immune system, especially macrophage-mediated innate immunity. In a previous study, we identified 1,25(OH)_2_D_3_-responsive and vitamin D receptor (VDR)-bound super-enhancer regions in THP-1 cells. Herein, we examined the transcriptional regulation of *ArfGAP with SH3 Domain, Ankyrin Repeat and PH Domain 2* (*ASAP2*) (encoding a GTPase-activating protein) by 1,25(OH)_2_D_3_ through the top-ranked VDR-bound super-enhancer region in the first intron of *ASAP2* and potential functions of ASAP2 in macrophages. First, we validated the upregulation of *ASAP2* by 1,25(OH)_2_D_3_ in both THP-1 cells and macrophages. Subsequently, we identified three regulatory regions (i.e., the core, 1,25(OH)_2_D_3_-responsive, and inhibitory regions) in the VDR bound-enhancer of *ASAP2*. ASAP2 promoted RAC1-activity and macrophage efferocytosis in vitro. Next, we assessed the functions of ASAP2 by mass spectrometry and RNA sequencing analyses. ASAP2 upregulated the expressions of antiviral-associated genes and interacted with SAM and HD domain-containing deoxynucleoside triphosphate triphosphohydrolase 1 (SAMHD1). In vivo, vitamin D reduced the number of apoptotic cells in experimental autoimmune encephalomyelitis (EAE) and promoted macrophage efferocytosis in peritonitis without changing the mRNA level of *ASAP2*. Thus, we could better understand the regulatory mechanism underlying *ASAP2* transcription and the function of ASAP2, which may serve as a potential treatment target against inflammatory diseases and virus infections.

## 1. Introduction

Vitamin D plays an important role in anti-inflammation partly by regulating the innate immune system, especially macrophages. For example, vitamin D can inhibit the production of pro-inflammatory cytokines in monocytes and macrophages by regulating mitogen-activated protein kinase phosphatase 1 [1]. Furthermore, it can downregulate IL-8 expression in hyperinflammatory macrophages, thus serving as a potential anti-inflammatory treatment strategy for chronic inflammatory lung disease [2]. 1,25(OH)_2_D_3_, the active form of vitamin D_3_, also promotes anti-infectious innate immune responses, such as upregulation of the expressions of antimicrobial peptides β-defensin 2 and cathelicidin antimicrobial peptide [3]. Vitamin D receptor (VDR), the only cognate receptor of the active form of vitamin D_3_ (i.e., 1,25(OH)_2_D_3_), as a transcription factor (TF), can bind multiple genomic regions, thus interacting with multiple other TFs and coregulators for transcriptional regulation [4].

Super-enhancers (SEs) are dense clusters of enhancers identified by chromatin accessibility assays (Formaldehyde-Assisted Isolation of Regulatory Elements-sequencing (FAIRE-seq) identified regions or DNaseI hypersensitive sites), pioneer/master TFs (e.g., PU.1 for monocytes and RORγt for Th17 cells), and pervasive factors in the transcription machinery (e.g., p300, MED1, BRD4, and RNA polymerase II). These can strongly promote gene transcription and predict key genes having cell-specific and stimulation-responsive functions [5,6]. The regulation of 1,25(OH)_2_D_3_ on its target genes is strictly dependent on spatio-temporal VDR binding sites (including 1,25(OH)_2_D_3_-driven super-enhancers) in the context of chromatin (e.g., cellular, temporal, and species specificity) [7,8,9]. In our previous study, we identified three genes (i.e., *DENN Domain Containing 6B* (*DENND6B), Ubiquitin Specific Peptidase 2 (USP2),* and *ArfGAP with SH3 Domain, Ankyrin Repeat and PH Domain 2 (ASAP2)*) showing high expressional levels and top-ranked type I VDR super-enhancers (i.e., the VDR super-enhancer (VSE) regions without pre-occupied VDR but transforming to VSE after 1,25(OH)_2_D_3_ stimulation) in THP-1 cells [6]. Although the intensity of *ASAP2* VSE is high enough for identification as a super-enhancer in a range of 12.5 kb using the rank ordering of super-enhancers (ROSE) algorithm, the full length of *ASAP2* VSE is only 1146 bp. Therefore, we speculated it as a classic VDR-bound enhancer exerting a strong effect on gene transcription.

ASAP2 protein is an ADP-ribosylation factor GTPase-activating protein (ARFGAP) that can activate downstream GTPase ARF6 and then RAC1, thus regulating vesicular transport and FcγR-mediated phagocytosis [10,11,12]. ASAP2 protein has multiple functional domains including the ARFGAP domain exhibiting GTPase-activating protein (GAP) activity. Owing to the persistent GTPase activities (i.e., ARF6 and RAC1), cycling between two forms—GTPase-GTP and GTPase-GDP—is necessitated. Both overexpression and mutation of *ASAP2* lead to the inhibition of the F-actin assembly at the phagocytosis cup [10,13,14]. Selenoprotein K-dependent palmitoylation of ASAP2 mediates the dissociation of ASAP2 from the phagocytic cup, crucial to the efficiency of FcγR-mediated phagocytosis [15]. Because actin assembly and cytoskeletal remodeling occur in both phagocytosis and efferocytosis, and the latter plays an important role in anti-inflammatory effects by preventing secondary necrosis and resultant inflammation [16], we hypothesized that 1,25(OH)_2_D_3_-upregulated ASAP2 expression could promote efferocytosis in THP-1 derived macrophages and attenuate tissue inflammation.

Therefore, herein, we examined the transcriptional regulatory mechanism underlying 1,25(OH)_2_D_3_ action on *ASAP2* via its VDR-bound enhancer region and confirmed the enhancing effect of 1,25(OH)_2_D_3_ and ASAP2 on efferocytosis both in vitro and in vivo. Furthermore, we assessed other functions of ASAP2 by mass spectrometry and RNA-sequencing (RNA-seq) analyses. We provided evidence for the association between 1,25(OH)_2_D_3_ and efferocytosis in an inflammatory disease model, which requires further validation in the future.

## 2. Material and Methods

### 2.1. Cells Culture and In Vitro Efferocytosis Assay

THP-1 cells (2 × 10^5^ per well) (Procell, Wuhan, China) in Roswell Park Memorial Institute (RPMI) 1640 (#350-000-CL, WISENT, Nanjing, China) supplemented with 1% 100× penicillin and streptomycin (#C002, Beyotime, Shanghai, China) and 10% fetal bovine serum (FBS, #086-150, WISENT, Nanjing, China) were seeded in a 24-well plate and stimulated with 50 ng/mL 12-O-tetradecanoyl phorbol-13-acetate (PMA, #N2060, APExBIO, Boston, USA). After 48 h, 20 ng/mL IL-4 (#CX03, Novoprotein, Suzhou China) and 20 ng/mL IL-13 (#CC89, Novoprotein, Suzhou, China) were added, and the cells were incubated for another 48 h to induce M2 polarization. IFN-γ (20 ng/mL, #C014, Novoprotein, Suzhou, China) and lipopolysaccharide (LPS, 100 ng/mL, #BS904, Biosharp, Hefei, China) were added to induce M1 polarization. After polarization, 100 nM 1,25(OH)_2_D_3_ (#B2141, APExBIO, Boston, USA) was added to the cells for another 48 h in the 1,25(OH)_2_D_3_ group before co-culturing with apoptotic cells.

J774A.1 cells (1 × 10^5^ per well) (Procell, Wuhan, China) in Dulbecco’s Modified Eagle Medium (DMEM, #319-005-CL, WISENT, Nanjing, China) supplemented with 1% 100× penicillin and streptomycin and 10% FBS were seeded in a 12-well plate.

PKH26 (#D0030, Solarbio, Beijing, China)-stained Jurkat cells (an immortalized line of human T lymphocyte cells, Procell, Wuhan, China) were exposed to UV irradiation (254 nm) for 15 min and cultured in RPMI 1640 without serum for 3 h at 37 °C, with 5% CO_2_ to induce apoptosis. Subsequently, apoptotic Jurkat cells were co-cultured with macrophages (1:5 for M2:Jurkat and 1:5 for J774A.1:Jurkat) for another 2 h (THP-1 M2) or 40 min (J774A.1), as indicated.

Peripheral blood mononuclear cells (PBMCs) were obtained from healthy adult donor buffy coats using Ficoll-Paque Plus (#17144002, Cytiva, Uppsala, Sweden) by gradient centrifugation following the manufacturer’s protocol. The experiment was approved by the Biomedical Ethics Committee of Anhui Medical University (No. 20190324). PBMCs were cultured in RPMI-1640 medium supplemented with 10% FBS and 1% 100× penicillin and streptomycin overnight. The next day, the medium was removed and replaced with fresh media. Subsequently, the cells were treated with fresh medium containing 20 ng/mL human M-CSF (#C417, Novoprotein, Suzhou, China) for 7 days. Macrophages were incubated in a fresh medium with or without 100 nM 1,25(OH)_2_D_3_ for 48 h.

### 2.2. Quantitative Real-Time Polymerase Chain Reaction (qRT-PCR)

Total RNA was extracted using the TRIzol reagent (#YY101, Epizyme, Shanghai, China) following the manufacturer’s protocols and reverse-transcribed into cDNA using Hifair^®^ III 1^st^ Strand cDNA Synthesis SuperMix (#11141ES60, YEASEN, Shanghai, China) for qRT-PCR. qRT-PCR was performed using Hieff^®^ qPCR SYBR Green Master Mix (#11201ES50, YEASEN, Shanghai, China) on the Lightcycler 96 real-time PCR system (Roche, Switzerland). The primer sequences used in this study are listed in Appendix A. Relative gene expression was determined by the 2^−ΔΔCt^ method. The expressional levels of genes were normalized to the expression of GAPDH, i.e., a “housekeeping” gene, as an endogenous reference. The primer sequences used in this study are shown in Appendix A.

### 2.3. Western Blot Analysis

Cells were lysed in radio-immunoprecipitation assay (RIPA) lysis buffer (#P0013B, Beyotime, Shanghai, China), and protein concentrations were determined using the BCA kit (#P0012, Beyotime, Shanghai, China). Proteins were separated by sodium dodecyl sulphate-polyacrylamide gel electrophoresis (SDS-PAGE) and transferred onto polyvinylidene difluoride (PVDF) membranes (Millipore, Billerica, MA, USA). After blocking, membranes were incubated with primary antibodies specific to ASAP2 (#sc-374323, Santa Cruz Biotechnology, CA, USA), VDR (#ab109234, ABCAM, Cambridge, UK), or GAPDH (#200306-7E4, ZENBIO, Chengdu, China) overnight at 4 °C. Subsequently, specific binding was detected using corresponding secondary antibodies (HRP-conjugated goat anti-rabbit IgG or HRP-conjugated goat anti-mouse IgG, Proteintech, Wuhan, China). Chemiluminescent signals were visualized using NcmECL High (#P2100, New Cell Molecular, Suzhou, China), and images were captured on a digital imaging system (Tanon, Shanghai, China).

### 2.4. Stable Knockdown of ASAP2

The shRNAs were chemically synthesized by Genechem, China. Oligos for shRNA targeting *ASAP2* (target sequence: shNC TTCTCCGAACGTGTCACGT; shASAP2-1 ccGGTGTCATTTGTGCACTTT, shASAP2-2 ccTGGATAAACAGACAGGGAA, and shASAP2-3 gcCTCAAACCTTCCATTGAAA) were inserted into the GV493 vector. Lentiviruses were prepared from 293T cells transfected with GV493 vector and packaging plasmid mix (Helper 1.0 and Helper 2.0, Genechem, Shanghai, China). THP-1 cells were seeded in a 12-well plate, and the lentiviral construct (LV-ASAP2-RNAi) was transfected. After culturing at 37 °C for 12–16 h, a fresh complete medium was added to the culture. Cells were selected in 2 μg/mL puromycin, 48 h after infection, for another 48 h. Cells were then cultured in fresh medium for the next experiment or RNA-seq analysis, whereby THP-1 cells were differentiated into macrophages by adding 50 ng/mL PMA for 48 h. Our RNA-seq data are in the GEO database with accession no. GSE271201.

### 2.5. Chromatin Immunoprecipitation (ChIP) Assay

The ChIP assay for VDR was performed using an Enzymatic Chromatin IP Kit (#9005, CST, MA, USA). The lysates of THP-1-derived macrophages were sheared by sonication and incubated with micrococcal nuclease to generate DNA fragments. Antibodies against VDR (#ab109234, ABCAM, Cambridge, UK) and normal rabbit IgG (#2729, CST, MA, USA) were used for immunoprecipitation. Normal rabbit IgG was used as the negative control. The ChIP products were quantified by PCR using specific primers (VSE: Forward Primer: GCCCTTCAGAAGATTCTGAAA, Reverse primer: CACACTAGCATTTGAAGTTCAC) followed by gel electrophoresis.

### 2.6. Luciferase Assay

The pGL4.23[luc2/minP] vector (#E8411, Promega, Madison, USA) containing different truncated VSE regions (pGL4.23-Basic, pGL4.23-VSE, pGL4.23-960, pGL4.23-640, and pGL4.23-420) was co-transfected with the pRL-TK vector (#E2241, Promega, USA) into THP-1 cells using lipofectamine TM 2000 (#11668019, Thermofisher, MA, USA) and nucleic acid transfection enhancer (NATE^TM^; #lyec-nate, Invivogen, San Diego, USA) following manufacturer’s instructions. Cells were harvested after 48 h and luciferase activities were measured on the Dual-Luciferase^®^ Reporter Assay System (#E1910, Promega, Madison, USA) following the manufacturer’s instructions. The firefly luciferase activity was normalized to that of renilla luciferase. Site-directed deletion mutagenesis was performed according to the instructions specified in the Fast Mutagenesis Kit (#4992901, Tiangen Biotech, Beijing, China). mut-VDR: GAACTT Deletion; mut-RUNX2: GGTGGT Deletion; mut-RUNX3: CTTCAA Deletion; mut1-CEBPA: TGGAGA Deletion, and mut2-CEBPA: GGATTG Deletion were generated.

### 2.7. Pulldown Assay for Testing RAC1 Activity

THP-1 monocytes were plated in a 10 cm Petri dish at the density of 5 × 10^6^ cells/mL and differentiated into macrophages after stimulation with 50 ng/mL PMA. After co-culturing with apoptotic Jurkat cells for 2 h, lysates of THP-1-derived macrophages were prepared using the Rac1 Activation Magnetic Beads Pulldown Assay kit (#17-10394, Merck Millipore, Darmstadt, German). Glutathione S transferase (GST) fused to the p21 binding domain of p21-activated kinase PAK1 (GST- PAK1 PDB), and glutathione magnetic beads in the kit were used for lysate incubation at 4 °C for pulling down RAC1-GTP after washing thrice with magnesium lysis buffer. The levels of RAC1-GTP in the pulldowns and the total RAC1 in the whole lysates were compared by Western blot analysis.

### 2.8. Immunoprecipitation Assay

Cells were incubated with immunoprecipitation lysis buffer on ice for 30 min. The cell lysates (1 mg total protein) were incubated with the indicated primary antibodies (2 µg, for VDR (#ab109234, ABCAM, Cambridge, UK), SAM, and HD domain-containing deoxynucleoside triphosphate triphosphohydrolase 1 (SAMHD1) (#12586-1-AP, Proteintech, Wuhan, China)) overnight at 4 °C. Subsequently, protein A/G beads (#sc-2003, Santa Cruz, CA, USA) were added to cell lysates and incubated at 4 °C for 6 h. Normal rabbit IgG (#A00002, ZENBIO, Chengdu, China) or Normal mouse IgG (#A00001, ZENBIO, Chengdu, China) was used as the negative control. After washing the beads, protein samples were analyzed by mass spectrometry or Western blot analysis.

### 2.9. Analysis of Differentially Expressed Genes (DEGs) and Functional Enrichment

The “GEOquery” and “limma” packages in R were used to assess the DEGs [17,18]. Significant DEGs were selected based on the following criteria: adjust *p* < 0.05 and |log_2_ fold change (FC)| > 1. Gene ontology (GO) analysis (including biological process (BP), molecular functions (MF), and cellular components (CC) terms) [19] and Kyoto Encyclopedia of Genes and Genomes (KEGG) pathways [20] were used to annotate the functions of genes via “clusterProfiler” R package [21]. Gene set enrichment analysis (GSEA) was performed by pre-ranking genes based on Log_2_FC [22]. GSEA analysis was performed using the c2.cp.reactome.v7.2.symbols.gmt (Reactome) gene set from MsigDB [22] via “clusterProfiler” R package.

### 2.10. The Disease Model of Experimental Autoimmune Encephalomyelitis (EAE) and Peritonitis

For EAE induction, after feeding on normal vitamin D and vitamin-D-deficient feed (Medicience, Yangzhou, China) for two weeks, 10-week-old female C57BL/6 mice were immunized subcutaneously (s.c.) with 200 μg MOG_35-55_ (MEVGWYRSP-FSRVVHLYRNGK) (#163913-87-9, ChinaPeptides, Shanghai, China) in complete Freund’s adjuvant (CFA) (#F5881, Sigma-Aldrich, St. Louis, USA) at four flank sites. Next, *Bordetella pertussis* toxin (PTX; #180243A1, List Biological Laboratories, Campbell, USA) was injected intraperitoneally on days 0 and 2. Body weights and clinical scores were assessed daily. Clinical scores of EAE were assessed according to the following criteria: 0, no disease; 1, limp tail; 2, hind-limb weakness; 3, partial hind-limb paralysis; 4, complete paralysis of one or more limbs; and 5, moribund state. The animal experiments were approved by the Animal Ethics Committee of Anhui Medical University (No. LLSC 20190338).

Peritonitis was induced by injection of 3% Brewer’s thioglycolate (#LA4590, Solarbio, Beijing, China) in the peritoneal cavity. After 48 h, sterile 100 mg 1,25(OH)_2_D_3_ in 200 μL phosphate buffer saline (PBS) was injected intraperitoneally per mouse in the 1,25(OH)_2_D_3_ group (200 μL sterile PBS was injected in the control group). Subsequently, 1 × 10^7^ PKH26-stained apoptotic thymocytes induced by 1 μmol/L dexamethasone for 6 h in RPMI supplemented with 10% FBS were injected intraperitoneally for the in vivo efferocytosis assay. Thymi were collected from 4–6-week-old female C57BL/6 mice. After red blood cells were lysed with red blood cell lysis buffer, thymocytes were washed twice and cultured (10^7^ cells/mL) in RPMI-1640 medium with 1% (*v*/*v*) 100× penicillin and streptomycin and 10% FBS.

### 2.11. Fluorescence-Activated Cell Sorting (FACS) Analysis

For EAE, mice were anesthetized with isoflurane and perfused with 10 mL PBS to remove any circulating red blood cells and leukocytes from the brain. Next, the brain was removed from the skull, homogenized in Hanks’ Balanced Salt Solution (HBSS) (#H1025, Solarbio, Beijing, China) containing 1 mg/mL collagenase A (#10103578001, Roche, Basel, Switzerland) and 1 mg/mL DNase I (#BS137, Biosharp, Beijing, China), and incubated at 37 °C for 20 min. After passing through a 70 μm mesh filter, mononuclear cells were separated from myelin by density gradient centrifugation (800 g for 10 min) over a 27% freshly prepared Percoll solution. Subsequently, these mononuclear cells were blocked with anti-mouse-CD16/32 (#101319, Biolegend, San Diego, USA) and stained with antibodies, including anti-mouse-CD45-PE (#103106, Biolegend, San Diego, USA), anti-mouse-CD45-PerCP/Cyanine5.5 (#103132, Biolegend, San Diego, USA), anti-mouse/human-CD11b-APC/Cyanine7 (#101226, Biolegend, San Diego, USA), and anti-mouse-F4/80-PerCP/Cyanine5.5 (#123127, Biolegend, San Diego, USA). Single cells were gated by FSC-A (area) versus FSC-H (height). Flow cytometry was performed on the cytometer (BeckmanCoulter, USA), and the data were analyzed using the FlowJo 10.4 software.

### 2.12. Histological Analysis and Immunofluorescence Staining

For EAE, the spinal cords were flushed with PBS from the spinal columns of sacrificed mice and fixed overnight in 4% paraformaldehyde (PFA, #BL539A, Biosharp, Hefei, China). Tissue sections were stained with hematoxylin and eosin (H&E) or Luxol fast blue (LFB) for evaluating inflammation and demyelination, respectively. Slides were visualized by light microscopy using a Panoramic tissue cell quantitative analyzer (TG, Austria). For assessment of apoptosis/efferocytosis in the spinal cord, detection of the apoptotic cells in tissues was performed using a Fluorescein (FITC) TUNEL Cell Apoptosis Detection Kit (#G1501, Service, Wuhan, China) following the manufacturer’s protocol. The nuclei were counterstained with DAPI (#MX4209, Maokang, Shanghai, China) for 5 min. Slides were visualized using a Nikon fluorescent microscope (Nikon, Japan).

### 2.13. Statistical Analyses

GraphPadPrism 7.0 was used for data analysis. Data are presented as mean ± SEM. Unpaired *t*-tests and multiple *t*-tests or one-way ANOVA followed by Dunnet’s post hoc correction were used for comparisons between two or multiple groups, respectively. Significance was indicated as follows: *, *p* < 0.05; **, *p* < 0.01, and ***, *p* < 0.001.

## 3. Results

### 3.1. ASAP2 Was Significantly Upregulated in THP-1 Cells and Macrophages following 1,25(OH)_2_D_3_ Exposure

To validate the upregulation of *ASAP2* in THP-1 following exposure to 1,25(OH)_2_D_3_, as suggested previously [23,24], we tested both mRNA and protein levels of *ASAP2* in THP-1 cells (Figure 1A,B). Given the activating effect of the GTPase-activating protein, ASAP2, on cytoskeletal-associated GTPases (i.e., ARF6 and RAC1) in macrophages [10,11,12], we tested the mRNA and protein levels of *ASAP2* in THP-1-derived macrophages with an active cytoskeletal rearrangement. Consistent with our hypothesis, both the mRNA and protein levels of *ASAP2* were significantly upregulated in macrophages with different polarization statuses (i.e., M0, M1, and M2) (Figure 1C,D). Moreover, the mRNA levels of *ASAP2* were significantly upregulated by 1,25(OH)_2_D_3_ in PBMC-derived macrophages (Figure 1E).

Because super-enhancer can regulate the transcription of multiple proximate genes within a chromatin loop [6], we also tested the mRNA level of *ITGB1BP1* located next to *ASAP2* (Figure 1F,G) and found a significant correlation between the genes before and after 1,25(OH)_2_D_3_ stimulation (Figure 1G).

### 3.2. Three Regulatory Regions in the VDR-Bound Enhancer Region of ASAP2 Regulated the Transcription of ASAP2 in Response to 1,25(OH)_2_D_3_

VSEs are dense clusters of enhancers identified by VDR binding regions. Our previous analysis suggested that [6] *ASAP2* VSE (GRCh37/hg19 chr2: 9400523-9401668) was a 1,25(OH)_2_D_3_ stimulation-dependent enhancer and could not be identified as a super-enhancer without 1,25(OH)_2_D_3_ stimulation, as evidenced by the ChIP-seq signal density of VDR binding in GSE89431 [25]. These VSEs in 1,25(OH)_2_D_3_-stimulated THP-1 cells were significantly enriched for the motifs of TFs VDR, RUNX3, and CEBPA [6]. Therefore, first, we validated VDR binding in the *ASAP2* VSE region by ChIP assay (Figure 2A). Subsequently, we searched the binding sites of the TFs (i.e., VDR, RUNX3, and CEBPA) in the *ASAP2* VSE region based on their motifs in the JASPAR database. Because RUNX1/2 and RUNX3 share similar motif sequences [26,27], we also searched for regions with RUNX1 and RUNX2 motifs. Interestingly, the top (i.e., the highest relative scores estimated by JASPAR) motif regions of VDR and RUNX3 were close to each other in the *ASAP2* VSE, and the top motif regions of SPI1 (the key and pioneer TF of THP-1) and CEBPA (the pioneer TF of THP-1) were close to each other in the region (Figure 2B). ChIP-seq data in GSE89431 [25] indicate that the VDR motif-enriched region between -640 and -420 and SPI1 motif-enriched region between -960 and -640 indeed show VDR ChIP-seq peak and SPI1 ChIP-seq peak, respectively (Figure 2B).

Based on the motif regions of VDR/RUNX3 and SPI1/CEBPA, we inserted full-length and three truncated versions of VSE into a pGL4.23-Basic luciferase reporter plasmid (Figure 2C). All the plasmids were transiently transfected into THP-1 cells. Luciferase assay revealed that the pGL4.23-640 fragment showed the highest transcriptional activity in THP-1 cells without 1,25(OH)_2_D_3_ stimulation, but the full-length pGL4.23-VSE had the highest activity following 100 nM of 1,25(OH)_2_D_3_ treatment for 48 h, indicating the stimulation (1,25(OH)_2_D_3_)-responsivity of *ASAP2* VSE, which may be attributed to the VSE/960 regions (i.e., the truncated region between pGL4.23-VSE and pGL4.23-960) given the significantly (*p* < 0.001) increased luciferase activity of pGL4.23-VSE in the 1,25(OH)_2_D_3_ group compared to the control group (Figure 2D). Moreover, the luciferase activity of pGL4.23-960 was lesser compared to that of pGL4.23-VSE in the 1,25(OH)_2_D_3_ group, while this reduction was not found in the control group (Figure 2D). Furthermore, in the 1,25(OH)_2_D_3_ group, the increased luciferase activity of pGL4.23-640 compared to that of pGL4.23-960 indicated the inhibitory effect of the 960/640 region (i.e., the truncated region of pGL4.23-640 compared to pGL4.23-960) (Figure 2D). However, without 1,25(OH)_2_D_3_ treatment the truncation of the 960/640 region from pGL4.23-960 did not enhance the luciferase activity of pGL4.23-640 (Figure 2D), suggesting that the inhibitory effect of 960/640 region was also 1,25(OH)_2_D_3_-dependent. Notably, compared to that of pGL4.23-640, the luciferase activity of pGL4.23-420 reduced significantly in both control and 1,25(OH)_2_D_3_ groups (Figure 2D), suggesting that the 640/420 region was a 1,25(OH)_2_D_3_-independent core enhancer region.

To validate the importance of VDR and RUNX3 binding sites in the 640/420 core enhancer region, these sites were separately deleted by site-directed mutagenesis (Figure 2E). All these deletions significantly reduced the corresponding luciferase activities of pGL4.23-640 (Figure 2F).

Notably, the 960/640 inhibitory region had SPI1/CEBPA motifs. We speculated that the inhibitory 960/640 region locked the activity of 1,25(OH)_2_D_3_-responsive VSE/960 region, which could be liberated by stimulation with 1,25(OH)_2_D_3_, resulting in the whole activation of the VSE. To examine the inhibitory effect of CEBPA binding, we deleted the two CEBPA binding sites separately (Figure 2G) and found that the first deletion of the CEBPA binding site enhanced the activity of VSE after 1,25(OH)_2_D_3_ treatment (Figure 2H).

### 3.3. 1,25(OH)_2_D_3_ Promoted M2 Macrophage Efferocytosis and Transcription of Certain Genes Encoding the “Eat-me” Signals

1,25(OH)_2_D_3_ has an anti-inflammatory effect on myeloid cells [1,4]. Efferocytosis is an anti-inflammatory process involving the clearance of apoptotic cells and post-engulfment anti-inflammatory signaling pathways [28]. The association between 1,25(OH)_2_D_3_ and efferocytosis remains unclear. M1 polarized macrophages are pro-inflammatory, while M2 macrophages resolve inflammation and show the “efferocytic-high” phenotype [29,30]. Notably, macrophage efferocytosis can further promote M2 polarization [31]. Herein, we co-cultured M2 macrophages with apoptotic Jurkat cells. 1,25(OH)_2_D_3_ could promote macrophage efferocytosis efficiently (i.e., the proportion of efferocytotic macrophages in all macrophages, Figure 3A,B) and increase the number of total attached apoptotic Jurkat cells (including the number of phagocytosed Jurkat cells (attached previously) and attached cells, Figure 3A,C).

Before the cytoskeletal-associated engulfing process, the “eat-me” signals should be identified. Considering the increased number of attached cells in 1,25(OH)_2_D_3_ group (Figure 3C), we tested the transcriptional level of genes encoding phosphatidylserine (PS) receptors and bridging factors for the “eat-me” signals before and after treatment with 100 nM 1,25(OH)_2_D_3_ for 48 h, including *TAM* (*TRYO3*, *AXL*, and *MERTK*), *BAI1*, *TIM4*, *ITGAV*, *ITGB3*, *ITGB5*, *GAS6*, *MFGE8,* and *STAB2*. Among them, *TIM4* and *MFGE8* were significantly upregulated in M1 and M0/M2 macrophages following 1,25(OH)_2_D_3_ treatment, respectively, and *BAI1* was upregulated modestly in all macrophage types (Figure 3D–F), apart from the unaffected or downregulated genes (Figure 3G–N).

### 3.4. 1,25(OH)_2_D_3_ Promoted Macrophage Efferocytosis Partly via ASAP2

ASAP2 is involved in vesicle transport and FcγR-mediated phagocytosis, as it activates ARF6 and, subsequently, ARF6-associated RAC1 [13,32]. However, whether ASAP2, following 1,25(OH)_2_D_3_ stimulation in macrophages, is involved in the process of efferocytosis by regulating cytoskeletal rearrangement remains unknown. Therefore, we knocked down *ASAP2* in THP-1 by shRNA (Figure 4A,B) and induced the THP-1-derived M2 macrophage by co-culturing with PKH26-stained apoptotic Jurkat cells (Figure 4C). The knockdown of *ASAP2* attenuated both the basal and 1,25(OH)_2_D_3_-enhanced macrophage efferocytosis (Figure 4C,D). Notably, shASAP2 did not reduce the efferocytosis to its basal level in the 1,25(OH)_2_D_3_ group (Figure 4D), suggesting other potential efferocytosis-associated and 1,25(OH)_2_D_3_-downstream molecules.

Because RAC1 activity mediates the cytoskeletal rearrangement for phagocytosis/efferocytosis [33,34,35], we tested RAC1-GTP levels after 1,25(OH)_2_D_3_ stimulation during efferocytosis. 1,25(OH)_2_D_3_ promoted the level of RAC1-GTP, but the knockdown of *ASAP2* reduced the level of RAC1-GTP (Figure 4E,F), suggesting the activating role of ASAP2 on RAC1 during efferocytosis.

RAC1, as a terminal effect GTPase of multiple activation pathways such as the STAT3-VAV1-RAC1 pathway, SRC-PI3K-RAC1 pathway, and ELMO1–DOCK180-RAC1 complex, is involved in actin cytoskeletal remodeling during efferocytosis [36,37,38]. To validate the role of RAC1 in 1,25(OH)_2_D_3_-enhanced efferocytosis, we used a RAC1-specific inhibitor, azathioprine (150 μM for 48 h), to block its activity and found significantly attenuated efferocytosis (Figure 4G,H).

### 3.5. The Cytoskeletal-Associated Proteins and Interactors of ASAP2

Furthermore, mass spectrometry was performed to identify the interactors of ASAP2 (Figure 5A,B, Appendix A) and enriched functions (Figure 5C,D) in 1,25(OH)_2_D_3_-treated macrophages.

After GO and KEGG enrichment analyses for 191 potential interactors, seven proteins, including SPECC1L/FKBP15 with the top score (Protein Q-score > 8) and the cytoskeletal-associated RAC1, were found to be enriched in the annotation of “actin filament” (Figure 5C), consistent with the above results.

By DAVID cluster analysis for the ASAP2 interactome, cytoskeleton-associated cluster 15 was identified, including six top-scoring ASAP2 interactors (i.e., EIF2S3, DNAJC13, SPECC1L, MPRIP, RHOC, and DOCK10) and the cytoskeletal-associated RAC1 (Figure 5D). EIF2S3, DNAJC13, SPECC1L, MPRIP, RHOC, DOCK10, FKBP15, and RAC1 are both ASAP2 interactors and cytoskeletal-associated proteins.

### 3.6. ASAP2 Promoted Interferon Signaling and Anti-Virus-Associated Pathways

To distinguish the functions of ASAP2 between 1,25(OH)_2_D_3_-stimulated THP-1 monocytes and 1,25(OH)_2_D_3_-stimulated THP-1 M0 macrophage, we performed mass spectrometry analysis in 1,25(OH)_2_D_3_-stimulated THP-1 cells (Appendix A) and obtained the overlap between the ASAP2 interactomes of THP-1 and THP-1 M0 cells (Figure 6A).

A total of 412 THP-1-specific interacting proteins mainly enriched in functions of “RNA splicing”/”Spliceosome” and “RNA transport” (Figure 6B) were found. A total of 150 THP-1 M0-specific interacting proteins enriched in multiple distinct functions, including “cell adhesion molecule binding”/”cadherin binding”, “intermediate filament cytoskeleton”/”intermediate filament”/”actin filament”, “mitochondrial inner membrane”/”mitochondrial protein complex”, “translation initiation”, and “RNA catabolic process” (Figure 6C) were identified, indicating the involvement of ASAP2 in the regulation of the cytoskeleton, mitochondrial function, and transcription/post transcription.

Notably, the cytoskeletal-associated functions (i.e., “intermediate filament cytoskeleton”, “intermediate filament”, and “actin filament”) were only enriched for the interactors in THP M0 macrophages. Furthermore, 41 common ASAP2 interactors between THP-1 and THP-1 M0 (Figure 6A) were identified and found to be enriched in the functions of RNA splicing, telomere maintenance/organization, and DNA metabolic process (Figure 6D). Interestingly, these 41 interacting proteins, including SAMHD1, were enriched in type I interferon-mediated signaling pathway, a result reported herein for the first time.

To examine the functions of ASAP2, RNA-seq analysis was performed for shASAP2 and shNC samples (Appendix A, GSE217201). Knockdown of *ASAP2* reduced the expression of genes involved in response to interferon (IFN) signaling and anti-virus-associated pathways (Figure 6E,F). By overlapping enriched genes in IFN signaling/anti-virus functions obtained from GO and GSEA enrichment analyses, 45 IFN signaling/anti-virus-associated genes (including *SAMHD1*) were identified and found to reduce following *ASAP2* knockdown (Figure 6G,H), suggesting the necessity of ASAP2 for IFN signaling/anti-virus-associated pathways. Interestingly, SAMHD1 was also enriched in the type I IFN-mediated signaling pathway as an ASAP2 interactor (Figure 6D). Therefore, we performed an immunoprecipitation assay for ASAP2 and SAMHD1 in 1,25(OH)_2_D_3_-treated THP-1-derived macrophages and validated their interaction (Figure 6I,J).

### 3.7. Vitamin D Reduced the Number of Apoptotic Cells in EAE and Promoted Macrophage Efferocytosis in Peritonitis without Changing the Total mRNA Level of Asap2

To examine the effect of vitamin D on efferocytosis and alteration of *Asap2* expression in inflammatory diseases, we constructed a model for central nervous system (CNS) inflammation EAE and thioglycolate peritonitis.

Although we observed the modestly (not significantly) protective effect of vitamin D on EAE given the relatively lower EAE score (Figure 7A) and higher body weight (Figure 7B) in the vitamin D normal group compared to the vitamin-D-deficient group, inflammatory cell infiltration (Figure 7C–E) and demyelination (Figure 7F) showed no difference between the two groups. Notably, an increased number of apoptotic cells in the spinal cord of the vitamin D deficient group was observed by TUNEL staining (Figure 7G). We also tested the mRNA level of *Asap2* in brain mononuclear cells, which showed no difference between vitamin D normal and -deficient groups (Figure 7H). After analyzing the microarray data of human brain samples (GSE131282), the mRNA level of *ASAP2* was found to increase in the lesions of gray matter from multiple sclerosis (MS) patients compared to the normal regions from MS patients or healthy individuals (Figure 7I). Based on Protein Atlas and UCSC Cell Browser on single-nucleus RNA-seq data of MS lesions [39], the transcription of *ASAP2* was mainly observed in the astrocytes (Figure 7J–K), suggesting the potential cell specificity of ASAP2 in CNS for future reference.

We assessed the protective effect of 1,25(OH)_2_D_3_ on peritonitis. Consistent with our hypothesis, although 1,25(OH)_2_D_3_ obviously reduced the number of recruited immune cells in the peritoneum, the peritoneal macrophages showed higher efferocytosis efficiency for the injected PKH26-stained apoptotic thymocytes in 1,25(OH)_2_D_3_ group (Figure 8A,B). To elucidate the role of Asap2 underlying 1,25(OH)_2_D_3_-strengthened efferocytosis, we tested the mRNA level of *Asap2* in all peritoneal immune cells before and after intraperitoneal (i.p.) injection of apoptotic thymocytes. The expressional level of *Asap2* reduced modestly (not significantly increased) following 1,25(OH)_2_D_3_ injection (Figure 8C,D), mostly due to its complex spatio-temporal expression in vivo.

## 4. Discussion

This study attempted to elucidate the regulatory mechanism of VDR-bound enhancer on the transcription of *ASAP2* in THP-1 cells and the potential effects of upregulated ASAP2 levels in THP-1-derived macrophages. We found three regulatory regions in the *ASAP2* VDR-bound enhancer, which separately played the role of core enhancer, inhibition, and promotion for the transcription of *ASAP2* after 1,25(OH)_2_D_3_ stimulation. 1,25(OH)_2_D_3_ could promote efferocytosis of apoptotic Jurkat cells by THP-1 M2 macrophages partly through the upregulation of *ASAP2* expression and activation of RAC1. The previously unknown functions of ASAP2, including RNA splicing, RNA transport, mitochondrial complex, and anti-virus, were revealed for the first time by mass spectrometry and RNA-seq analyses. Both EAE and peritonitis models validated the promotive effect of vitamin D on efferocytosis in vivo, while the total mRNA level of ASAP2 remained unchanged. Our findings indicated the regulatory mechanism and potential functions of ASAP2, suggesting its potential as a treatment target against inflammatory diseases.

The direct and indirect regulation of vitamin D on its target genes has been extensively studied, suggesting a highly cell-specific manner of action, a significant overlap of biological process regulation in humans and mice, and the critical spatio-temporal vitamin D-driven super-enhancer regions in the context of chromatin [7,8,9]. VDR super-enhancer is a stretch of VDR binding regions with a high rank of VDR ChIP-seq signal intensity, potentially showing a strong enhancer activity for functional genes in specific cell types. In addition to VDR, other transcription factors (i.e., RUNX3 and CEBPA) are enriched in the VDR super-enhancer regions, thus cooperating with VDR and regulating gene transcription. The RUNX family interacts with VDR in different cell types [26,27]. Herein, we showed that RUNX3 cooperated with VDR to upregulate the transcription of *ASAP2* at the core enhancer and promoting regions in THP-1 cells. SPI1 (the key/pioneer TF of THP-1 [40,41]) and CEBPA (the pioneer TF for THP-1 [24,42]) cooperate to inhibit the transcription of *ASAP2* at the inhibitory region of the VDR bound-enhancer. The mechanism of their inhibition on 1,25(OH)_2_D_3_-upregulated *ASAP2* may be due to the changed spatial structure of the transcription loop and warrants further investigation.

ARFGAP ASAP2 is associated with FcγR-mediated phagocytosis and vesicular transport. For the first time, we showed that the interactome of ASAP2 in THP-1 and the common interacting proteins between THP-1 and THP-1 M0 that were enriched in the RNA and DNA regulatory functions, e.g., RNA splicing, telomere maintenance, DNA metabolic process, and the type I interferon-mediated signaling pathway. Only in THP-1 M0, the ASAP2 interacting proteins were enriched in actin-associated functions. Among these functions, ASAP2 not only interacted with proteins in the IFN signaling pathway but also regulated the expression of genes in the IFN signaling pathway and antiviral immunity. SAMHD1, a dNTP triphosphohydrolase (dNTPase), can hydrolyze intracellular dNTPs, thereby reducing viral reverse transcription and cDNA synthesis [43,44]. Vitamin D can also promote antibacterial and antiviral innate immunity by inducing antimicrobial peptides, lowering intracellular iron concentration, and enhancing autophagy [45,46]. Previously, the association between IFN signaling/anti-virus immunity and other ARFGAPs has been identified. For example, ARFGAP Domain-Containing Protein 2 (ADAP2) can mediate IFN responses, promote type I IFN production, and restrict the entry of RNA viruses [47,48].

Because *ASAP2* is a risk gene associated with MS risk SNP rs1109670 located upstream [49], we first tested the effect of vitamin D on the EAE model. However, herein, we did not find a significant protective effect of normal vitamin D on CNS inflammation, which was potentially due to the attenuated antigen-presenting ability within peripheral lymph nodes in the vitamin D deficient group. Although the positive correlation between vitamin D status and MS and the protective effect of vitamin D on EAE mice has been confirmed [4,50], some studies show lower or unchanged EAE clinical scores in the vitamin-D-deficient group compared to the normal group, potentially due to the weakened antigen-presenting ability in peripheral lymph nodes [51,52,53,54]. Herein, we showed the modestly protective effect of vitamin D on EAE and its potential promoting effect on efferocytosis due to the fewer apoptotic cells in the spinal cord of the normal vitamin D group, suggesting the potentially important role of efferocytosis in MS. However, we did not observe upregulated *ASAP2* levels in the vitamin D normal group and needs further validation in specific cell types (macrophage/microglia/astrocytes) or pathological sections (normal appearing/lesion regions) of the CNS in the future.

The inhibitory effect of vitamin D on peritoneal dialysis-related peritonitis, bacterial peritonitis, and zymosan-induced peritonitis has been reported previously [55,56,57] given its regulatory ability on innate immune cells by increasing anti-infectious effect and resolving inflammation. Moreover, upregulated IL-10 levels in regulatory T (Treg) cells can enhance efferocytosis and further attenuate peritoneal inflammation [36]. Herein, we showed that active vitamin D_3_ could increase the efferocytosis of macrophages in thioglycolate-induced peritonitis. However, we did not observe any alterations in *ASAP2*, suggesting stable expression of *ASAP2* or potential time-dependent upregulation during peritonitis.

In summary, we identified different regulatory regions of VDR-bound enhancer in *ASAP2* and the promotive effect of 1,25(OH)_2_D_3_/ASAP2/RAC1 axis on efferocytosis, thus suggesting its potential as a new therapeutic target against inflammatory diseases. Several new functions of ASAP2 in macrophages were identified by mass spectrometry and RNA-seq analyses. The interaction between ASAP2 and SAMHD1 in the regulation of the IFN signaling pathway and antiviral immunity needs further studies for validation as a potential antiviral target.

## Figures and Tables

**Figure 1 nutrients-14-04935-f001:**
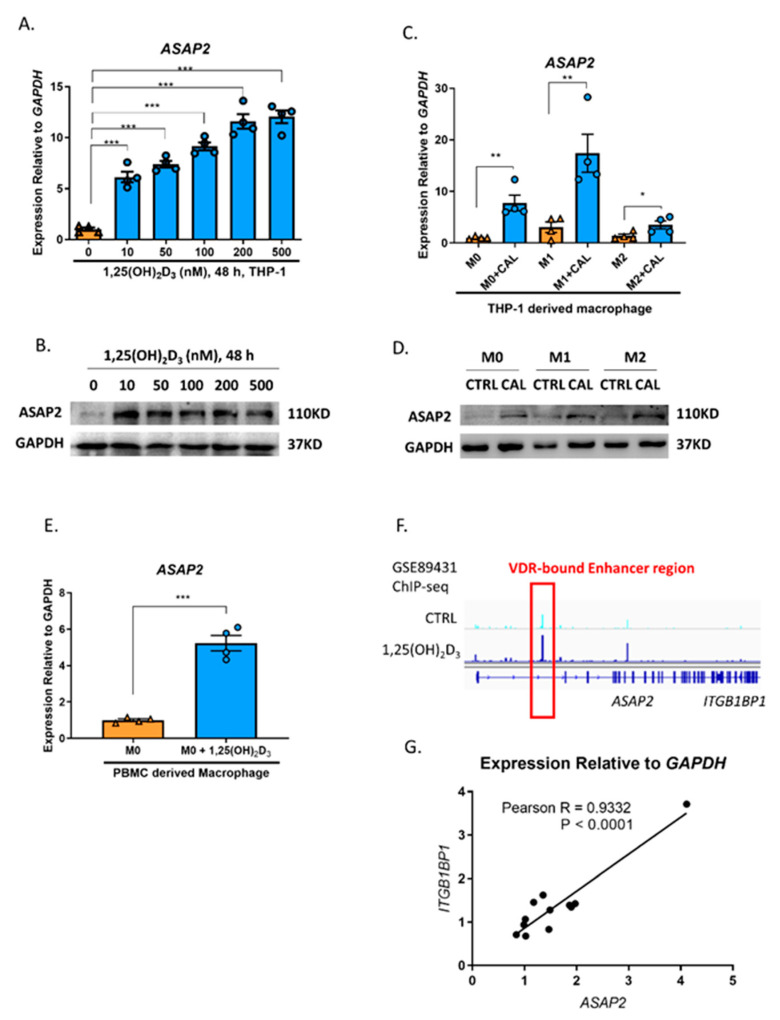
The promotive effect of 1,25(OH)_2_D_3_ on the expression of *ASAP2*. (**A**,**B**). The transcriptional (**A**) and expressional (**B**) levels of *ASAP2* in THP-1 cells following treatment with different concentrations of 1,25(OH)_2_D_3_. (**C**,**D**). The transcriptional (**C**) and expressional (**D**) levels of *ASAP2* in M0/M1/M2 macrophages following 100 nM 1,25(OH)_2_D_3_ treatment for 48 h. (**E**) The transcriptional level of *ASAP2* following 100 nM 1,25(OH)_2_D_3_ for 48 h in M0 macrophages. (**F**) The genomic binding density of VDR in *ASAP2/ITGB1BP1* and the *ASAP2* enhancer region from GSE89431 ChIP-seq data (**G**). The positive correlation between mRNA levels of *ASAP2* and *ITGB1BP1*. CTRL, control; CAL, 1,25(OH)_2_D_3_. Data are representative of four independent experiments for A/C/E, three independent experiments for B/D and two independent experiments for (**G**). Error bars show means ± SEM. Unpaired *t*-tests were performed in A/C/E and Pearson’s correlation analysis in (**G**). *, *p* < 0.05; **, *p* < 0.01; ***, *p* < 0.001.

**Figure 2 nutrients-14-04935-f002:**
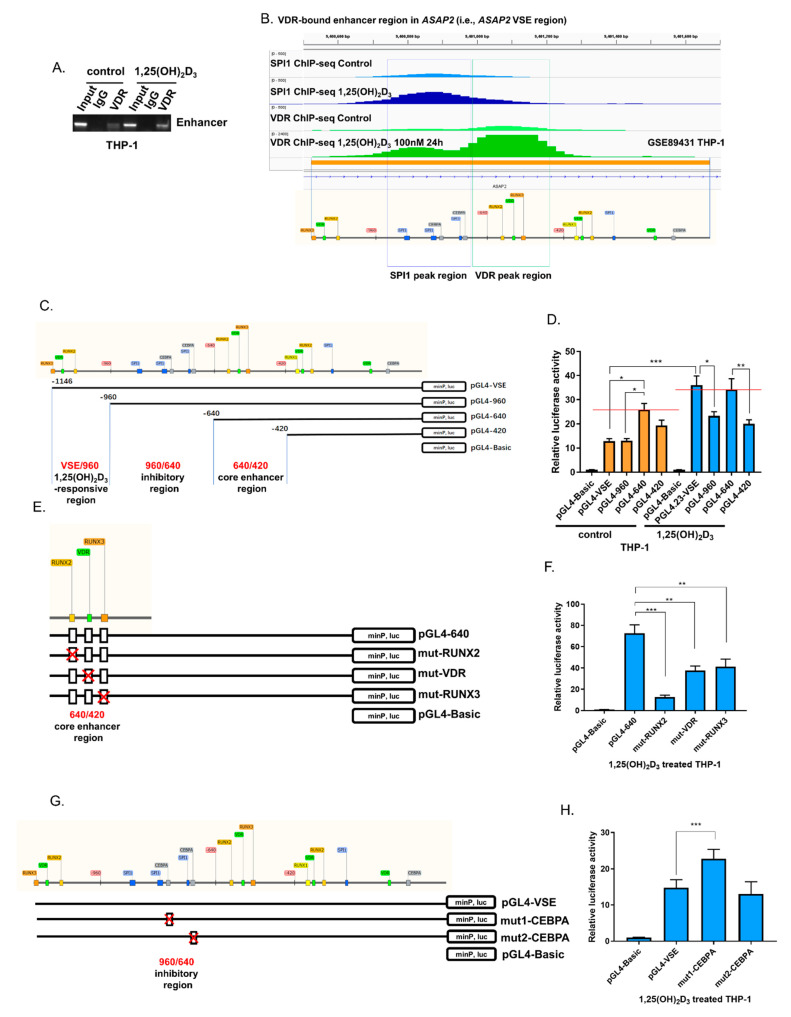
1,25(OH)_2_D_3_ upregulates ASAP2 via the VDR bound-enhancer region. (**A**) ChIP assay using VDR antibody on the VSE region of *ASAP2.* (**B**) The genomic binding density (peak) of SPI1 and VDR in the *ASAP2* VSE region from GSE89431 ChIP-seq data, with the corresponding genomic region showing motif positions of transcription factors. (**C**) The schematic illustration for the construction of luciferase reporter pGL4.23 plasmids with full-length VSE and four truncated fragments. (**D**) Cells were harvested 48 h after transfection, and enhancer activities were measured by a luciferase assay. (**E**) The schematic illustration for different pGL4.23-640 deletion mutants. (**F**) The enhancer activities for different deletion mutants were measured by luciferase assay. (**G**) The schematic illustration for different VSE deletion mutants. (**H**) The enhancer activities for different deletion mutants were measured by luciferase assay. Data are representative of four independent experiments for D/F/K and three independent experiments for G/I. One-way ANOVA followed by Dunnett’s post hoc test was used for calculating the statistical significance. Data are representative of four independent experiments in D/F/H. VSE, VDR super-enhancer. *, *p* < 0.05; **, *p* < 0.01; ***, *p* < 0.001.

**Figure 3 nutrients-14-04935-f003:**
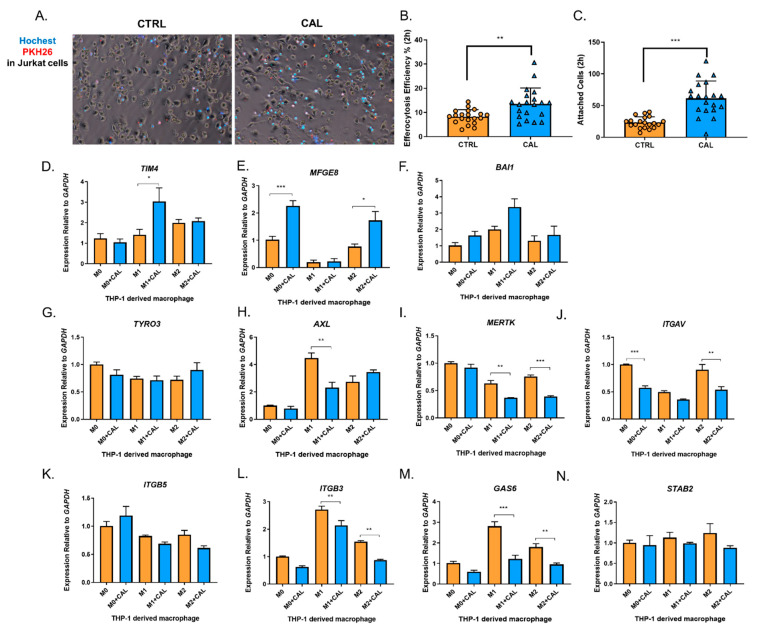
1,25(OH)_2_D_3_ promoted macrophage efferocytosis and upregulated certain efferocytosis-associated genes. (**A**) Representative images of macrophage efferocytosis for apoptotic Jurkat cells from three independent experiments. (**B**–**C**) Efferocytosis efficiency (**B**) and attached apoptotic cells (**C**) between CTRL (control) and CAL (1,25(OH)_2_D_3_) groups. (**D**–**N**). The regulation of 1,25(OH)_2_D_3_ on the transcription of efferocytosis-associated genes in M0/M1/M2 macrophages was assessed from 4–6 independent experiments. One-way ANOVA followed by Dunnett’s post hoc test was used for calculating the statistical significance. CTRL, control; CAL, 1,25(OH)_2_D_3_. *, *p* < 0.05; **, *p* < 0.01; ***, *p* < 0.001.

**Figure 4 nutrients-14-04935-f004:**
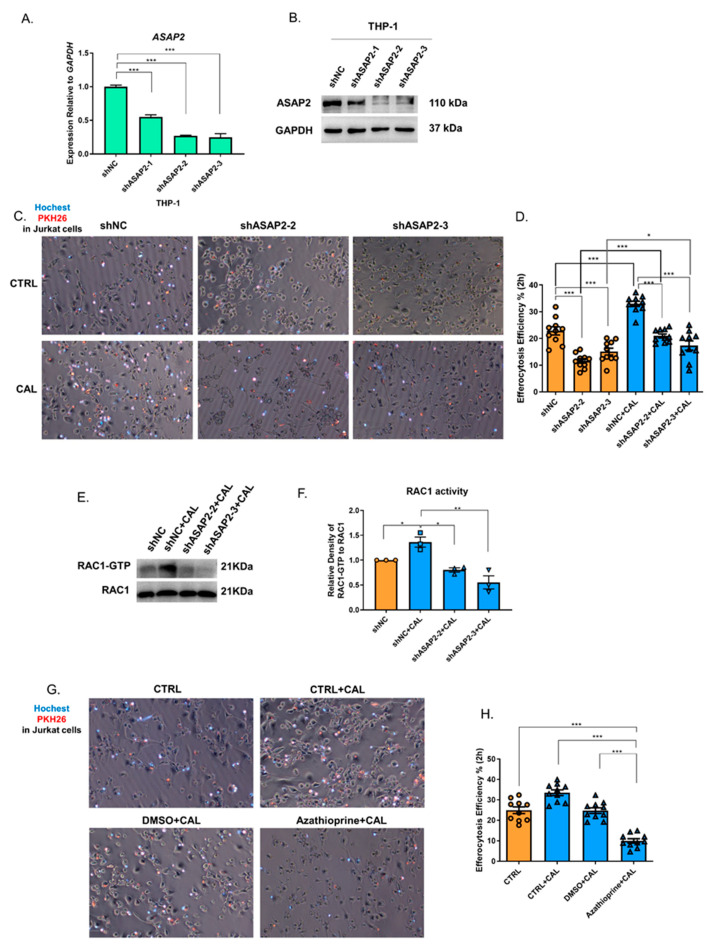
1,25(OH)_2_D_3_ promoted macrophage efferocytosis via ASAP2 and RAC1 activity. A–B. Validation of *ASAP2* knockdown with shASAP2 by RT-qPCR (**A**) and Western blotting (**B**) in THP-1 cells. (**C**) Representative images of macrophage efferocytosis of THP-1-M2 macrophages from three independent experiments. (**D**) The downregulated efferocytosis efficiency following *ASAP2* knockdown in THP-1-M2 macrophages. (**E**) Representative RAC1 pulldown assay in THP-1-M0 macrophage. (**F**) Relative densitometry value of RAC1-GTP to RAC1. (**G**) Representative images of macrophage efferocytosis of THP-1 M2 macrophages from three independent experiments. (**H**) The downregulated efferocytosis efficiency following RAC1 inhibitor, azathioprine, treatment in THP-1 M2 macrophages. CTRL, control; CAL, 1,25(OH)_2_D_3_. *, *p* < 0.05; **, *p* < 0.01; ***, *p* < 0.001.

**Figure 5 nutrients-14-04935-f005:**
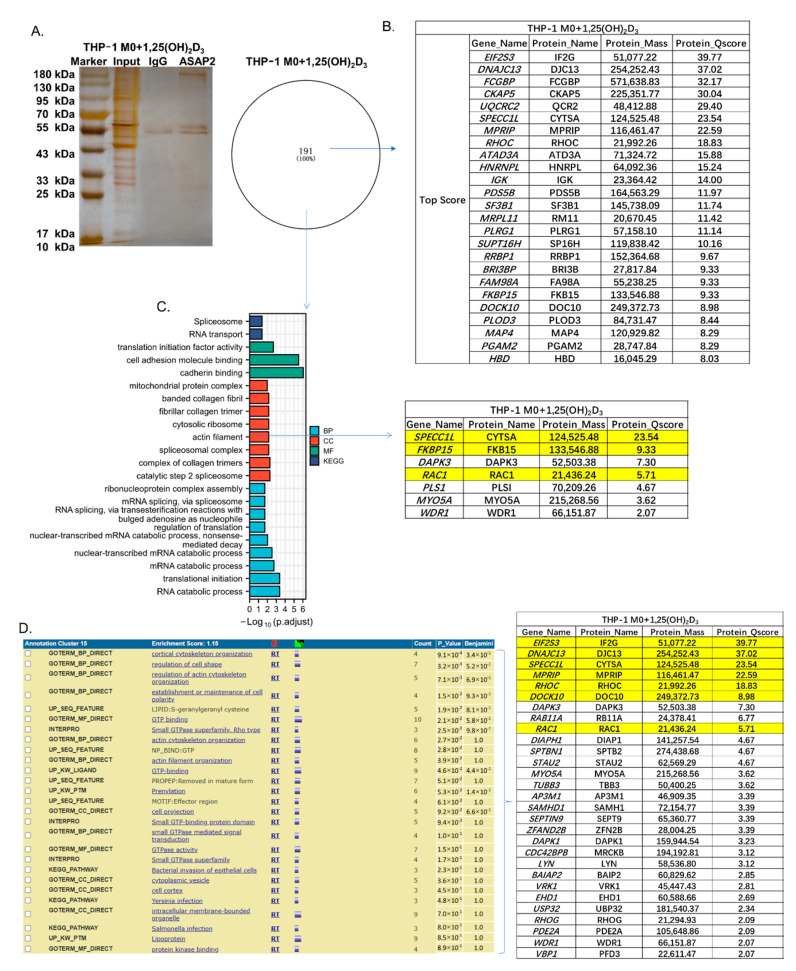
The interactome of ASAP2 in 1,25(OH)_2_D_3_-stimulated THP-1 M0 macrophages. (**A**) Silver staining analysis for ASAP2 interactors by immunoprecipitation with anti-ASAP2 antibody in 1,25(OH)_2_D_3_-stimulated THP-1 M0 macrophages. (**B**) The potential ASAP2 interactors with the top scores (Protein Q-score > 8). (**C**) The functions of ASAP2 interactors by GO and KEGG enrichment analyses and the enriched proteins in the annotation “actin filament”, including the yellow labeled top-scoring proteins and RAC1. (**D**) The enriched cytoskeleton-associated functional cluster 15 of ASAP2 interactors by DAVID cluster analysis and the enriched proteins, including the yellow labeled top-scoring proteins and RAC1.

**Figure 6 nutrients-14-04935-f006:**
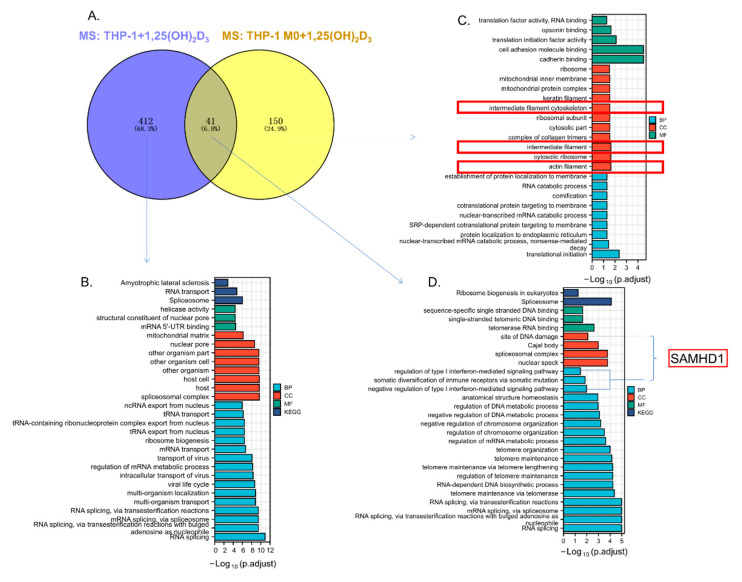
The potential antiviral function of ASAP2 based on the interaction with SAMHD1. (**A**) The Venn diagram represents the overlap between the interactome of ASAP2 in 1,25(OH)_2_D_3_-stimulated THP-1 monocytes and THP-1 M0 macrophages. (**B**) The enriched functions of THP-1 monocyte-specific ASAP2 interactors. (**C**) The enriched functions of THP-1 M0 macrophage-specific ASAP2 interactors. (**D**) The enriched functions of overlapped ASAP2 interactors between THP-1 and THP-1 M0. (**E**) The IFN signaling/anti-virus annotations enriched for ASAP2-upregulated genes (GO enrichment analysis). (**F**) The IFN signaling/anti-virus annotations enriched for ASAP2-upregulated genes (GSEA analysis). (**G**) Venn diagram representing the overlap between (**E**) and (**F**). (**H**) Heatmap showing the expressional level of the overlapping and IFN signaling/anti-virus associated genes between shNC and shASAP2 THP-1 M0 macrophages. (**I**,**J**) Interactions between ASAP2 and SAMHD1 were detected by co-immunoprecipitation (co-IP) with anti-ASAP2 (**I**) or anti-SAMHD (**J**) antibodies in 1,25(OH)_2_D_3_-stimulated THP-1 M0 macrophages. The immunoglobulin G (IgG) group was the negative control.

**Figure 7 nutrients-14-04935-f007:**
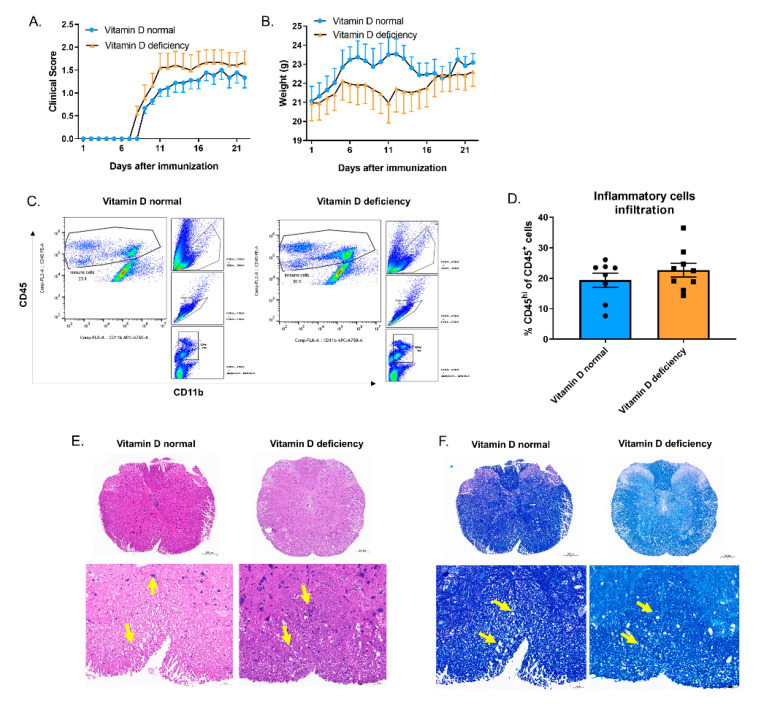
Vitamin D deficiency increased the number of CNS apoptotic cells. (**A**,**B**) Mean clinical scores (**A**) and weights (**B**) of age-matched female vitamin D normal and deficient mice subjected to MOG35-55-induced EAE (*n* = 9 mice per group in three repeats). (**C**,**D**) Flow cytometric analysis of immune cells in the brain (including CD45^+^ lymphocytes and myeloid cells and excluding microglia cells) (**C**) shows no significant difference between the vitamin D normal (*n* = 8 mice) and -deficient groups (*n* = 9 mice) in three repeats (**D**). (**E**,**F**) H&E staining (**E**) and LFB staining (**F**) of spinal cord sections from EAE-induced mice show inflammatory cell infiltration and demyelination, respectively (yellow arrows: E: inflammatory cell infiltration; F: demyelination). Scale bars: 200 µm (upper) and 50 µm (lower). (**G**) TUNEL staining for spinal cord sections from EAE-induced mice. (**H**) The transcriptional level of *Asap2* in CNS mononuclear cells. (**I**) The transcriptional levels of *ASAP2* between healthy gray matter (CTRL_GM), normal-appearing gray matter in MS patients (MS_NAGM), and lesioned gray matter in MS patients (MS_lesionGM) from the GSE131282 microarray dataset. (**J**) The transcriptional levels of *ASAP2* in brain cell subsets from Protein Atlas. (**K**) The transcriptional levels of *ASAP2* in brain cell subsets from UCSC Cell Browser (Range setting > 2.0). An unpaired *t*-test was performed in D/H/I.

**Figure 8 nutrients-14-04935-f008:**
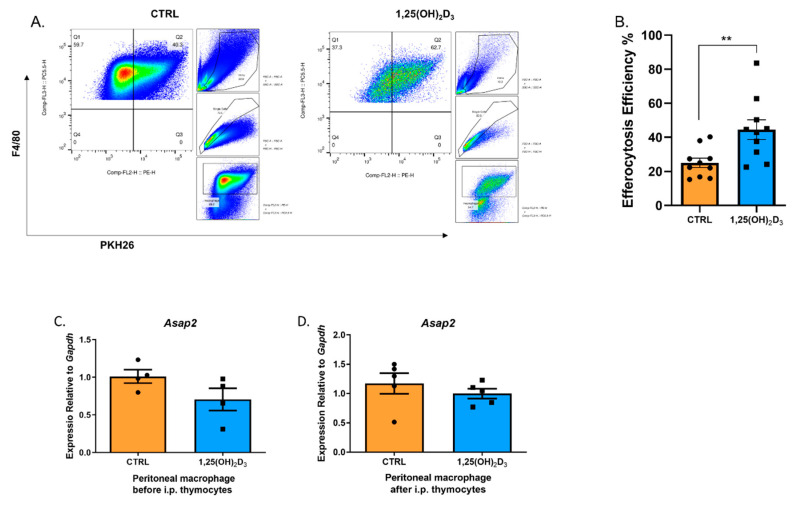
1,25(OH)_2_D_3_ increased efferocytosis of the peritoneal macrophages. (**A**,**B**) Flow cytometric analysis for peritoneal macrophages (**A**), showing a significantly enhanced efferocytosis efficiency (**B**) (*n* = 10 mice per group in three repeats). (**C**,**D**) The transcriptional levels of *Asap2* between control and 1,25(OH)_2_D_3_ groups before (**C**) (*n* = 4 mice per group in two repeats) and after (**D**) (*n* = 5 mice per group in two repeats) intraperitoneally injecting apoptotic thymocytes. An unpaired *t*-test was performed in B/C/D. **, *p* < 0.01.

## Data Availability

Datasets were analyzed in this study can be found in Appendix A and https://www.ncbi.nlm.nih.gov/geo/query/acc.cgi?acc=GSE217201 (GEO datasets) (accessed on 8 November 2022).

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
