# Peer review of "1,25(OH)_2_D_3_ Promotes Macrophage Efferocytosis Partly by Upregulating *ASAP2* Transcription via the VDR-Bound Enhancer Region and ASAP2 May Affect Antiviral Immunity"

_nutrients, 2022, doi:10.3390/nu14224935_

Round 1
Reviewer 1 Report
Major comments:
1. Please be consistent with vitamin D terminology and use only one name per compound, such as vitamin D3, 25(OH)D3 and 1,25(OH)2D3. Moreover, please be more specific who using the term "vitamin D", which compound is meant. Moreover, please retain from the abbreviation "VD" or "VD3" for vitamin D.
2. Genome-wide data need to be deposited on public repositories, such as GEO. Please provide an accession number and a reviewer's password, in order to inspect them. Moreover, more information about the quality of the data (e.g. MA and/or MDS plots) need to be provided.
3. All figures should have the same style concerning font size and style. Please make sure that in printed form all figures are readable.
4. Super-enhancers of vitamin D target genes in THP-1 have been extensively studied by other groups, such as those from Carlberg and White. Please cite and discuss their results more extensively.
Minor comments:
1. All abbreviations should be defined at their first time use and then consistently applied. This applies also to the Abstract.
2. Only the abbreviations of gene names should be in italic. Please have gene name abbreviations always in italic. This applies also to tables and figures.
3. Please use for all gene and protein names consistently the latest HuGO nomenclature.
Reviewer 2 Report
My only real issue is it’s not a superenhancer. These are defined by a wider range of biochemical parameters (and analsyed for example with the ROSE algorithm). The region identified is ~ 1100 bp and I get the sense it’s just a regulator enhancer.
So, the authors just need to admit they’ve carefuly characterized an enhancer region which is somewhat impactful, OR go back and justify this region by H3K27ac ChIP-Seq (H3K27ac isn’t mentioned in the manuscript)
So, overall it’s careful biochemistry of modest biological significance
Reviewer 3 Report
The manuscript entitled: „1,25(OH)2D3 promotes macrophage efferocytosis partly by upregulating ASAP2 transcription via the VDR super-enhancer region, and the potential effect of ASAP2 on antiviral immunity” presents the regulatory mechanism of VDR super-enhancer on the transcription of ASAP2 in THP-1 cells and the potential effects of the upregulated ASAP2 in THP-1 derived macrophages.
In my opinion, the work is very valuable. Many tests have been done, the results are well described and documented.
Questions:
1. Is there any animal testing permisson?
2. Subsection 2.1. – why 100 nM of 1,25(OH)2D3 was added? Why not different amount?
Author Response
- Is there any animal testing permisson?
Response: Yes, the animal experiment was permitted by Animal Ethics Committee of Anhui Medical University with code: LLSC 20190338.
2.Subsection 2.1. – why 100 nM of 1,25(OH)2D3 was added? Why not different amount?
Response: Yes, it’s a bit arbitrary. Because the VSE in ASAP2 were identified by ROSE algorithm on the VDR ChIP-seq data from GSE89431 (in which THP-1 cells were treated for 24 h with 100 nM 1,25(OH)2D3), and there was no significant difference among 10-500 nM groups (the order of magnitude) in our results, we chose 100 nM of 1,25(OH)2D3 for further experiment.
Round 2
Reviewer 1 Report
1. The authors should respect the format of the journal.
2. All changes should be highlighted, this applies also to abbreviations and references.
3. The authors should respect the nomenclature for genes. Only gene name abbreviations should be in italic.
4. Please write vitamin D3, when you mean it and not vitamin D. Moreover, 1,25(OH)2D3 should be used instead of calcitriol.
5. Without an GSE annotation from GEO the paper cannot be published.
6. All abbreviations should be defined at their first time use, this applies also to the Abstract.
Author Response
Comments and Suggestions for Authors
- The authors should respect the format of the journal.
Yes, we checked the format requirement and adjusted our files.
- All changes should be highlighted, this applies also to abbreviations and references.
Yes, according to editor’s suggestion, we marked up all revisions using the “Track Changes” in MS Word.
- The authors should respect the nomenclature for genes. Only gene name abbreviations should be in italic.
Yes, we rechecked our manuscript again and corrected them.
- Please write vitamin D3, when you mean it and not vitamin D. Moreover, 1,25(OH)2D3 should be used instead of calcitriol.
Yes, we used ‘vitamin D3’ instead of ‘vitamin D’ when we mean vitamin D3. We replaced all ‘calcitriol’ with ‘1,25(OH)2D3’ in the manuscript. We kept some ‘CAL’ in figures due to the limited space and gave necessary explanation in figure legend (‘CAL: 1,25(OH)2D3’).
- Without an GSE annotation from GEO the paper cannot be published.
Yes, our RNA-seq dataset has been checked by GEO and got the accession number: GSE271201 (https://www.ncbi.nlm.nih.gov/geo/query/acc.cgi?acc=GSE217201).
- All abbreviations should be defined at their first time use, this applies also to the Abstract.
Yes, we rechecked all the abbreviations, added necessary definitions at first time use, and abbreviated the words appearing later to keep consistency.
also Please see the attachment